# A *Plasmodium falciparum* MORC protein complex modulates epigenetic control of gene expression through interaction with heterochromatin

Maneesh Kumar Singh[1†], Victoria Ann Bonnell[2,3,4†], Israel Tojal Da Silva[5], Verônica Feijoli Santiago[6], Miriam Santos Moraes[1], Jack Adderley[7], Christian Doerig[7], Giuseppe Palmisano[6], Manuel Llinás[2,3,4,8], Celia RS Garcia[1*]

[1]Department of Clinical and Toxicological Analyses, School of Pharmaceutical Sciences, University of São Paulo, São Paulo, Brazil; [2]Department of Biochemistry and Molecular Biology, Pennsylvania State University, University Park, Harrisburg, United States; [3]Huck Institutes Center for Eukaryotic Gene Regulation, Pennsylvania State University, University Park, Harrisburg, United States; [4]Huck Institutes Center for Malaria Research, Pennsylvania State University, University Park, Harrisburg, United States; [5]Hospital AC Camargo, Centro Internacional de Pesquisa, São Paulo, Brazil; [6]Department of Parasitology, Institute of Biomedical Science, University of São Paulo, São Paulo, Brazil; [7]School of Health and Biomedical Sciences, RMIT University, Bundoora, Australia; [8]Department of Chemistry, Pennsylvania State University, University Park, Harrisburg, United States

*For correspondence:
cgarcia@usp.br

†These authors contributed equally to this work

**Abstract** Dynamic control of gene expression is critical for blood stage development of malaria parasites. Here, we used multi-omic analyses to investigate transcriptional regulation by the chromatin-associated microrchidia protein, MORC, during asexual blood stage development of the human malaria parasite *Plasmodium falciparum*. We show that *Pf*MORC (PF3D7_1468100) interacts with a suite of nuclear proteins, including APETALA2 (ApiAP2) transcription factors (*Pf*AP2-G5, *Pf*AP2-O5, *Pf*AP2-I, PF3D7_0420300, PF3D7_0613800, PF3D7_1107800, and PF3D7_1239200), a DNA helicase DS60 (PF3D7_1227100), and other chromatin remodelers (*Pf*CHD1 and *Pf*EELM2). Transcriptomic analysis of *Pf*MORC[HA-glmS] knockdown parasites revealed 163 differentially expressed genes belonging to hypervariable multigene families, along with upregulation of genes mostly involved in host cell invasion. In vivo genome-wide chromatin occupancy analysis during both trophozoite and schizont stages of development demonstrates that *Pf*MORC is recruited to repressed, multigene families, including the *var* genes in subtelomeric chromosomal regions. Collectively, we find that *Pf*MORC is found in chromatin complexes that play a role in the epigenetic control of asexual blood stage transcriptional regulation and chromatin organization.

## eLife assessment

This study provides **valuable** insights into how chromatin-bound *Pf*MORC controls gene expression in the asexual blood stage of *Plasmodium falciparum*. By interacting with key nuclear proteins, *Pf*MORC is predicted to affect expression of genes relating to host invasion and variable subtelomeric gene families. Correlating transcriptomic data with in vivo chromatin analysis, the study provides **convincing** evidence for the role of *Pf*MORC in epigenetic transcriptional regulation.

## Introduction

Despite global efforts to combat malaria, the disease caused an estimated 249 million new cases and more than 608,000 deaths in 2022 (World Malaria Report 2023). The etiological agents of human malaria are apicomplexan parasites of the genus *Plasmodium*. Among the six known *Plasmodium* species that can infect humans, *Plasmodium falciparum* is the most lethal, causing the majority of annual deaths (*Cowman et al., 2012*). The intraerythrocytic developmental cycle (IDC) of *P. falciparum* is responsible for the clinical symptoms of malaria. The IDC commences when merozoites generated during the liver stage enter the circulatory system and invade red blood cells (RBCs). While growing inside RBCs, parasites undergo multiple developmental phases with distinct morphological characteristics (ring, trophozoite, and schizont). Maturation and schizogony lead to the formation of 16–32 daughter merozoites destined to invade new RBCs (*Singh et al., 2010*). A small fraction (<10%) of parasites differentiate into non-replicative sexual gametocytes, which are transmitted to the mosquito during a second blood meal to complete sexual development.

Parasite development through the asexual blood stage is facilitated by precise transcriptional regulation, where genes are only expressed when needed in a just-in-time fashion (*Bozdech et al., 2003*). Approximately 90% of genes across the *P. falciparum* genome are transcribed during the asexual blood stage in a cascade of gene expression believed to be controlled by both sequence-specific transcription factors and dynamic epigenetic changes to chromatin (*Painter et al., 2011*; *Toenhake et al., 2018*; *Jeninga et al., 2019*; *Hollin et al., 2021*). The gene regulatory toolbox in malaria parasites is lacking many transcriptional regulatory factors conserved in other eukaryotes (*Gardner et al., 2002*). The identification of a large family of *Plasmodium* homologs of the plant APETALA2/Ethylene Response Factor (AP2/ERF) transcription factors (TFs) provided a breakthrough to unravel key regulators of gene expression in these parasites (*Balaji et al., 2005*). There are 28 putative APETALA2 (ApiAP2) TFs identified in *P. falciparum*, each protein containing 1–3 AP2 DNA binding domains (*Painter et al., 2011*; *Jeninga et al., 2019*). Multiple studies in *Plasmodium* spp., using a variety of approaches, have characterized essential functions of ApiAP2 TFs in RBC invasion, gametocytogenesis, oocyst formation, and sporozoite formation (*Yuda et al., 2009*; *Kafsack et al., 2014*; *Sinha et al., 2014*; *Modrzynska et al., 2017*; *Santos et al., 2017*; *Zhang et al., 2018*). ApiAP2 proteins in *Plasmodium* species display a wide array of functions during parasite development in both the vertebrate host and the mosquito vector, but surprisingly, over 70% of ApiAP2 proteins are expressed in the asexual blood stage of the lifecycle (*Bozdech et al., 2003*; *Le Roch et al., 2003*; *Chappell et al., 2020*).

Despite increasing evidence detailing ApiAP2 protein regulatory complexes (*Santos et al., 2017*; *Harris et al., 2019*; *Hillier et al., 2019*; *Hoeijmakers et al., 2019*; *Farhat et al., 2020*; *Josling et al., 2020*; *Miao et al., 2021*; *Srivastava et al., 2023*; *Yuda et al., 2023*; *Antunes et al., 2024*), the functional properties and specific interaction partners of many ApiAP2 TFs remain to be elucidated. A quantitative mass spectrometry-based analysis of the parasite protein interaction network has revealed links between ApiAP2 TFs and many chromatin remodelers, such as an extended Egl-27 and MTA1 homology 2 (EELM2) domain-containing proteins (PF3D7_0519800 and PF3D7_1141800), histone deacetylase protein 1 (HDAC1; PF3D7_0925700), and the microchidia family protein *Pf*MORC (PF3D7_1468100) (*Hillier et al., 2019*). Genome-wide mutagenesis studies revealed that several genes within these proposed networks, including *pfmorc*, are essential for parasite proliferation (*Bushell et al., 2017*; *Zhang et al., 2018*). Using a targeted deletion strategy, we previously were unable to delete *pfmorc*, further suggesting that it is essential for parasite growth (*Singh et al., 2021a*). Moreover, STRING network analysis has shown that the putative ApiAP2:*Pf*MORC complex forms a network with proteins having DNA-binding or nucleosome assembly properties, suggesting that *Pf*MORC may function as an accessory protein in epigenetic regulation (*Hillier et al., 2019*). Previous work identified *Pf*MORC in a chromatin complex containing the chromatin remodeling protein *Pf*SWI (PF3D7_0624600) located at *var* gene promoter regions (*Bryant et al., 2020*). All *P. falciparum* strains encode roughly 60 highly polymorphic *var* genes, but through an epigenetic allelic exclusion mechanism, each parasite is thought to expresses a single allele (*Real et al., 2022*; *Schneider et al., 2023*). Bryant et al. have proposed that *Pf*MORC is recruited to these heterochromatic regions to assist in the silencing of *var* genes, on the basis of its known function as a repressor complex component in model eukaryotes. In the related apicomplexan parasite *Toxoplasma gondii*, *Tg*MORC functions as a repressor of sex-associated genes by recruiting the *Tg*HDAC3 histone deacetylase and forming

heterogenous complexes with 11 ApiAP2 TFs (*Farhat et al., 2020*). To date, only two *Tg*MORC:HDAC3 complexes have been characterized, including a dimeric *Tg*AP2XII-2:HDAC3 complex and a heterotrimeric *Tg*AP2XII-1:AP2XI-2:HDAC3 complex (*Srivastava et al., 2023*; *Antunes et al., 2024*).

MORC proteins canonically consist of two major conserved regions: (1) a catalytic ATPase domain at the N-terminus (comprising a GHKL [Gyrase, HSP90, Histidine kinase, and MutL] domain and S5 fold domain) and (2) a C-terminal protein-protein interaction domain containing one or more coiled-coils (*Koch et al., 2017*). The conserved MORC gene family is present in most eukaryotes (with the exception of fungi), often with multiple paralogs per genome (*Dong et al., 2018*), and has been extensively investigated in various model systems in the context of epigenetic gene regulation. In plants, MORC proteins function in gene repression and heterochromatin compaction (*Koch et al., 2017*; *Zhong et al., 2023*). Additionally, MORC proteins have been shown to play diverse roles in metazoans by forming protein–protein interactions with immune-responsive proteins, SWI chromatin remodeling complexes, histone deacetylases, and histone tail modifications (*Iyer et al., 2008b*, *Kang et al., 2012*; *Moissiard et al., 2012*; *Bordiya et al., 2016*; *Kim et al., 2019*).

While most metazoans possess 5–7 MORC paralogs, apicomplexan parasites contain a single *morc* gene (*Iyer et al., 2008a*), which encodes not only the canonical animal-like GHKL ATPase domain, but also three Kelch-repeats, and a CW-type zinc-finger domain not found in mammalian MORCs (*Farhat et al., 2020*). This unique domain architecture suggests that the apicomplexan MORC proteins may have parasite-specific roles. To dissect the functional roles of *Pf*MORC, we conducted a proteomic analysis with *Pf*MORC[GFP], which identified several nuclear proteins, including a cluster of ApiAP2 TFs, as possible interacting partners. We also determined the genome-wide localization of *Pf*MORC at multiple developmental stages, which revealed *Pf*MORC recruitment predominantly to subtelomeric regions, corroborating that *Pf*MORC may act as a repressor of the clonally variant gene families that are important contributors to malaria pathogenesis. Finally, we performed transcriptomic analysis in *Pf*MORC[HA-glmS] knockdown parasites at the asexual stage to investigate alterations in global gene expression. We observed an overrepresentation of downregulated genes belonging to the heterochromatin-associated hypervariable gene family proteins. Overall, this study allows us to assign a role for *Pf*MORC in facilitating the plasticity of epigenetic regulation during asexual blood stage development.

## Results

### Proteins that co-purify with PMORC represent gene regulatory and chromatin remodeling components

A previous study using Blue-Native PAGE identified a *Pf*MORC complex in association with ApiAP2 proteins and chromatin remodeling machinery (*Hillier et al., 2019*). To validate this observation and expand the repertoire of *Pf*MORC interactors, we used a targeted immunoprecipitation approach coupled to LC–MS/MS proteomic quantification. We used a previously generated *Pf*MORC[GFP] parasite line (*Singh et al., 2021a*) to carry out immunoprecipitation with an anti-GFP antibody at the trophozoite stage, where *Pf*MORC is abundant (*Singh et al., 2021b*). The *P. falciparum* 3D7 strain expressing wild-type *pfmorc* was used a negative control. Trophozoite lysates were incubated with anti-GFP-Trap-A beads (ChromoTek, gta-20), and the immunocaptured proteins were resolved by SDS–PAGE (*Figure 1—figure supplement 1A*). We applied a label-free quantitative proteomics approach with a false discovery rate (FDR) of 1% and number of peptides ≥2 to excised gel samples to identify proteins interacting with *Pf*MORC[GFP]. From three biological replicates, we identified 211, 617, and 656 proteins, respectively. We further identified the overlap between all three wild-type 3D7 and *Pf*MORC[GFP] replicates and found a total of 132 and 142 proteins, respectively (*Figure 1—figure supplement 1B–D*, *Supplementary file 1*).

To analyze the relative ratio of proteins between wild-type 3D7 and *Pf*MORC[GFP] groups, we used the mean-normalized MS/MS count to calculate a fold change from *Pf*MORC[GFP]/3D7, and selected differentially abundant proteins above a 1.5× cutoff filter. This high stringency threshold was used to preclude any mis-identification of *Pf*MORC interactors caused by variability between the replicates (*Figure 1A*, *Figure 1—figure supplement 1D*). This analysis resulted in 143 significantly enriched proteins (*Supplementary file 2*). From these candidate *Pf*MORC-interacting proteins, the top enriched protein (20.8-fold enrichment) was *Pf*EELM2 (PF3D7_0519800, -log$_{10}$ p-value 3.43). *Pf*EELM2

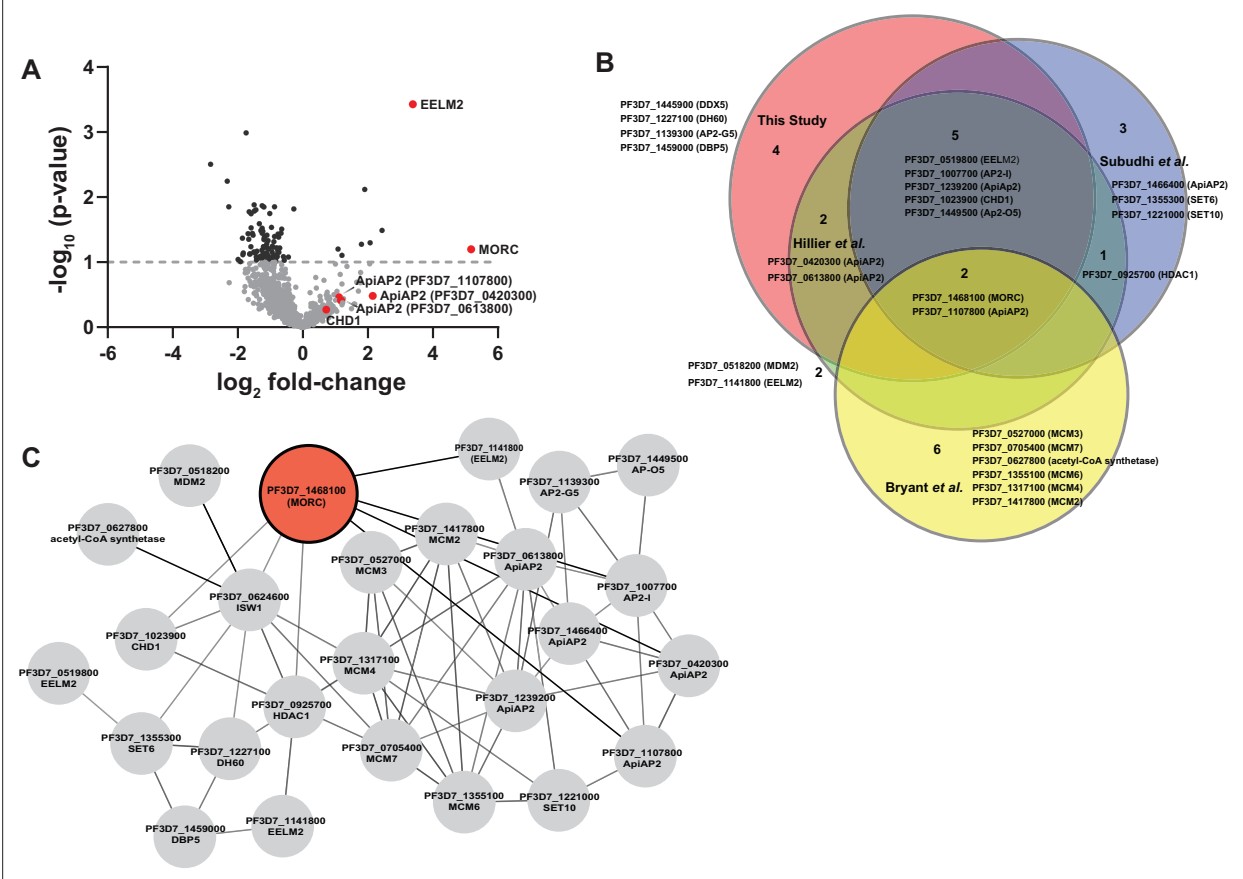

**Figure 1.** Proteomic analysis of parasites expressing *Pf*MORC^GFP reveals *Pf*MORC association with nuclear proteins of epigenetic regulation. (**A**) Volcano plot illustrates the protein enrichment in label free LC-MS/MS analysis of *Pf*MORC CoIPed proteins from three independent experiments at 32 hours post invasion (hpi). For normalized MS/MS counts, Student's *t*-test was performed and proteins were ranked as -$\log_2$ fold-change (x-axis) versus statistical p-values (y-axis). Gray dashed horizontal line shows the p-value cutoff. (**B**) Comparative analysis showing the juxtaposition of specific proteins CoIPed in *Pf*MORC^GFP with selected proteins from recent works of Hillier et al., Bryant et al., and Subudhi et al., where ApiAP2 or ISW1 were used as bait in similar CoIP experiments. The Venn diagram illustrates the overlap between identified proteins, revealing that the intersecting proteins are primarily ApiAP2 and chromatin remodelers. (**C**) An interactive protein–protein interaction network is constructed with proteins known to interact with *Pf*MORC, using proteins identified in this study and proteins documented in previously published works. Proteins identified in this study with known interaction networks from the STRING database were used to curate the network employing Cytoscape to enrich the network quality.

The online version of this article includes the following source data and figure supplement(s) for figure 1:

**Figure supplement 1.** Proteomic analysis of PfMORC interacting proteins identified in *Plasmodium falciparum* lysate.

**Figure supplement 1—source data 1.** Uncropped and labeled SDSPAGE gel.

**Figure supplement 1—source data 2.** Raw unedited SDSPAGE gel.

was previously predicted as a *Pf*MORC interactor (***Hillier et al., 2019***) and identified in a quantitative histone peptide pulldown to be consistently recruited to H2B.Z_K13/14/18a (***Hoeijmakers et al., 2019***). Similarly, EELM2 of the related Apicomplexan parasite *T. gondii* was recently identified in a *Tg*MORC-associated complex (***Farhat et al., 2020***). We also detected numerous ApiAP2 transcription factors (*Pf*AP2-G5, *Pf*AP2-O5, *Pf*AP2-I, PF3D7_1107800, PF3D7_0613800, PF3D7_0420300, and PF3D7_1239200) (***Table 1***), similar to results reported both by ***Hillier et al., 2019*** and in the *Toxoplasma* studies which also predicted or experimentally identified many ApiAP2 interactions (***Farhat et al., 2020***; ***Srivastava et al., 2023***; ***Antunes et al., 2024***). *Pf*AP2-G5 (PF3D7_1139300, -$\log_{10}$ p-value 0.22) and *Pf*AP2-O5 (PF3D7_1449500, -$\log_{10}$ p-value 0.41) were enriched 20.5-fold and 14.99-fold, respectively, suggesting that these factors are likely major components in complex with *Pf*MORC. To corroborate our results, we compared our *Pf*MORC^GFP coIPed proteins to a recently published, computationally predicted, protein–protein interaction network (***Hillier et al., 2019***; ***Bryant et al., 2020***; ***Subudhi et al., 2023***) and found many of ApiAP2 and EELM2 proteins shared across both

**Table 1.** Potential *Pf*MORC interacting proteins enriched in CoIP eluates and identified in LC-MS/MS from three independent experiments and fold change ≥1.5× GFP/3D7.

| Protein ID | Annotation | Fold change | -log p-value | Known function |
|---|---|---|---|---|
| PF3D7_0519800 | EELM2 domain-containing protein | 20.79 | 3.45 | - |
| PF3D7_1139300 | AP2 domain transcription factor AP2-G5 | 20.5 | 0.22 | Repressor of commitment and early gametocyte development (*Shang et al., 2021a*) |
| PF3D7_1449500 | AP2 domain transcription factor AP2-O5 | 14.99 | 0.41 | Regulator of mature ookinete motility (*Modrzynska et al., 2017*) |
| PF3D7_1468100 | *Pf*MORC | 11.76 | 1.20 | |
| PF3D7_1023900 | SNF2 helicase, putative or Chromodomain-helicase-DNA-binding protein 1 homolog, CHD1 | 10.61 | 0.30 | |
| PF3D7_1459000 | ATP-dependent RNA helicase DBP5 | 10.24 | 0.46 | |
| PF3D7_1227100 | DNA helicase 60, DH60 | 6.55 | 0.41 | - |
| PF3D7_1007700 | AP2 domain transcription factor AP2-I | 4.65 | 0.08 | Invasion (*Santos et al., 2017*; *Josling et al., 2020*) |
| PF3D7_1107800 | AP2 domain transcription factor | 3.11 | 0.36 | Master regulator of parasite growth, chromatin structure, and *var* gene expression (*Subudhi et al., 2023*) |
| PF3D7_0613800 | AP2 domain transcription factor | 2.43 | 0.28 | - |
| PF3D7_0420300 | AP2 domain transcription factor (ApiAP2) | 2.38 | 0.48 | - |
| PF3D7_0624600 | SNF2 helicase, ISW1 | 2.09 | 0.001 | *var* gene expression (*Bryant et al., 2020*) |
| PF3D7_1239200 | AP2 domain transcription factor | 2.01 | 0.25 | - |

datasets (*Figure 1B*). Collectively, our results demonstrate a direct association of *Pf*MORC with various chromatin-associated factors, including at least seven ApiAP2 proteins.

The potential interactors of *Pf*MORC that we detected in this experiment also included proteins implicated in DNA replication and repair, including the ATP-dependent RNA helicase DBP5 (PF3D7_1459000, -log$_{10}$ p-value 0.46) and the DNA helicase 60 DH60 (PF3D7_1227100, -log$_{10}$ p-value 0.41). *Pf*DH60 exhibits DNA and RNA unwinding activities, and its high expression in the trophozoites suggests a role in DNA replication (*Pradhan et al., 2005*). We also identified two putative chromatin-associated proteins, chromodomain-helicase-DNA-binding protein 1 CHD1 (PF3D7_1023900, -log$_{10}$ p-value 0.30) and the SNF2 chromatin-remodeling ATPase ISWI (PF3D7_0624600, -log$_{10}$ p-value 0.001), which are associated with chromosome structure maintenance, DNA replication, DNA repair, and transcription regulation. *Pf*ISWI was previously reported to be associated with *Pf*MORC in the context of *var* gene transcriptional activation during ring stage development (*Bryant et al., 2020*). Gene Ontology (GO) analysis was performed to identify enriched biological processes, cellular components, and molecular functions using a p-value cutoff of 0.05. We found significant enrichment of DNA-binding transcription factor activity and mRNA binding, transcription, and regulation of transcription activity (*Figure 1—figure supplement 1E*, *Supplementary file 3*). Overall, we again find that *Pf*MORC forms a link between ApiAP2 TFs and chromatin remodelers (*Figure 1C*).

### *Pf*MORC localizes to multigene families in subtelomeric regions

To determine the genome-wide occupancy of the *Pf*MORC chromatin-associated remodeling complex, we used chromatin immunoprecipitation followed by high-throughput sequencing (ChIP-seq). Using purified, crosslinked nuclear extracts, we immunoprecipitated *Pf*MORC from a 3xHA-tagged *Pf*MORC[HA-glmS] parasite line (*Singh et al., 2021a*) for ChIP-seq at the trophozoite stage (30 hpi) and the schizont stage (40 hpi) in biological duplicates. These timepoints represent the stages at which *Pf*MORC expression is the highest (*Singh et al., 2021b*). An independent ChIP-seq experiment in biological duplicate using anti-GFP and a *Pf*MORC[GFP] parasite line (*Singh et al., 2021b*) at the schizont stage was used to confirm our findings, demonstrating that the protein tags do not affect *Pf*MORC immunoprecipitation or genome-wide localization (*Figure 2—figure supplement 1A and B*). As an additional control, we correlated one no-epitope (3D7 wild type), negative control sample using the same anti-HA antibody on unmodified parasite lines for immunoprecipitation (*Figure 2—figure*

*supplement 1A and B*; *Bonnell et al., 2023*), which resulted in an expected low correlation to the tagged samples. The biological ChIP-seq replicates showed high fold enrichment (Log$_2$[IP/Input]) (*Figure 2—figure supplement 1C*) and were highly correlated with each other within each timepoint (*Figure 2—figure supplement 1D–F*).

We identified *Pf*MORC localized to subtelomeric regions on all chromosomes across the *P. falciparum* genome, with additional occupancy at internal heterochromatic islands (*Figure 2A and B*). Within the subtelomeric regions, *Pf*MORC was bound upstream and within the gene bodies of many hypervariable multigene families (*Figure 2C*), including *var* genes (*Figure 2—figure supplement 2*), *rif* genes (*Figure 2—figure supplement 3*), and *stevor* genes (*Figure 2—figure supplement 4*). Predicted binding sites (across both biological replicates) between the 30 hpi and 40 hpi timepoints showed a high degree of overlap, suggesting that *Pf*MORC binds many of the same regions throughout the later stages of asexual development when *Pf*MORC is highly expressed (*Figure 2D*). The proportion of *Pf*MORC-bound regions was similar across the 5′ upstream region of genes and the gene bodies throughout the genome, including subtelomeric regions (*Figure 2E*). As opposed to the binding of other proteins at the subtelomeric regions, such as the heterochromatin protein 1 (*Pf*HP1) (*Flueck et al., 2010*), *Pf*MORC occupancy is not widespread. Instead, it forms sharp peaks within, and adjacent to, HP1-bound regions (*Figure 2F*), suggesting a unique role for *Pf*MORC in heterochromatin condensation, boundary demarcation, and gene repression.

We further defined *Pf*MORC putative gene targets as genes displaying peaks within 2 kb upstream of the ATG start codon or within gene bodies. For those peaks between gene targets in a head-to-head orientation, the closest gene was chosen. This resulted in 149 putative gene targets at 30 hpi and 102 gene targets at 40 hpi. A close examination of the 84 overlapping genes shows that many are *var* genes, rRNA genes, and genes encoding exported proteins (*Supplementary file 4*), with GO terms related to cell adhesion, host–pathogen interactions, and antigenic variation (*Supplementary file 5*). The 65 uniquely bound genes at the 30 hpi timepoint showed an enrichment of highly expressed tRNA and rRNAs genes, as well as conserved unknown genes, while those 18 unique to the 40 hpi timepoint included a variety of late-stage expressed genes. Transcript abundance (*Chappell et al., 2020*) of the predicted *Pf*MORC gene targets at both the 30 hpi and 40 hpi timepoints form two major clusters: cluster 1 being genes expressed at the late ring/early trophozoite stage (10–24 hpi) and cluster 2 at the late schizont stage (40–48 hpi) (*Figure 2—figure supplement 5*). This two-cluster gene target pattern of expression mirrors the biphasic pattern of expression by *Pf*MORC, suggesting that *Pf*MORC could have distinct functions, forming complexes with different sets of transcriptional regulators, at various times during asexual proliferation. As determined in other eukaryotic organisms, MORC family proteins do not generally bind DNA in a sequence-specific manner; it is, therefore, likely that *Pf*MORC is recruited to these genome-wide regions by sequence-specific transcription factors, such as the ApiAP2 proteins identified in our proteomics results.

## Binding sites of *Pf*MORC overlap with ApiAP2 proteins and epigenetic marks

*Pf*MORC has previously been found to interact with several ApiAP2 proteins (*Hillier et al., 2019*; *Bryant et al., 2020*; *Singh et al., 2021b*), as does the *Toxoplasma* ortholog (*Farhat et al., 2020*; *Srivastava et al., 2023*; *Antunes et al., 2024*). We identified a clear overlap between genome-wide *Pf*MORC binding and putatively interacting ApiAP2 proteins using available ChIP-seq datasets. In addition to our protein–protein interaction results (*Table 1*), previous studies have also suggested that *Pf*MORC interacts with a broad array of ApiAP2 TFs, such as *Pf*AP2-G5, *Pf*AP2-O5, *Pf*AP2-I, PF3D7_1107800, PF3D7_0613800, PF3D7_0420300, and PF3D7_1239200 (*Hillier et al., 2019*; *Bryant et al., 2020*; *Subudhi et al., 2023*). Therefore, we compared binding sites between interacting ApiAP2s and *Pf*MORC using available ChIP-seq data on *Pf*AP2-G5, *Pf*AP2-O5, *Pf*AP2-I, PF3D7_1107800, PF3D7_0613800, and PF3D7_1239200 (*Josling et al., 2020*; *Shang et al., 2021b*, *Shang et al., 2022*). Interestingly, there is a degree of overlap between the binding sites of all six ApiAP2 TFs and *Pf*MORC, suggesting that *Pf*MORC and these ApiAP2 TFs may cooperate in the regulation of gene expression at these loci (*Figure 3A and B*, *Figure 3—figure supplement 1*). However, the available data cannot differentiate whether all these factors are in one complex together, form multiple smaller heterologous complexes, or are components of separate complexes in individual cells. DNA motif enrichment analysis (*Bailey, 2021*) identified several unique and significant DNA

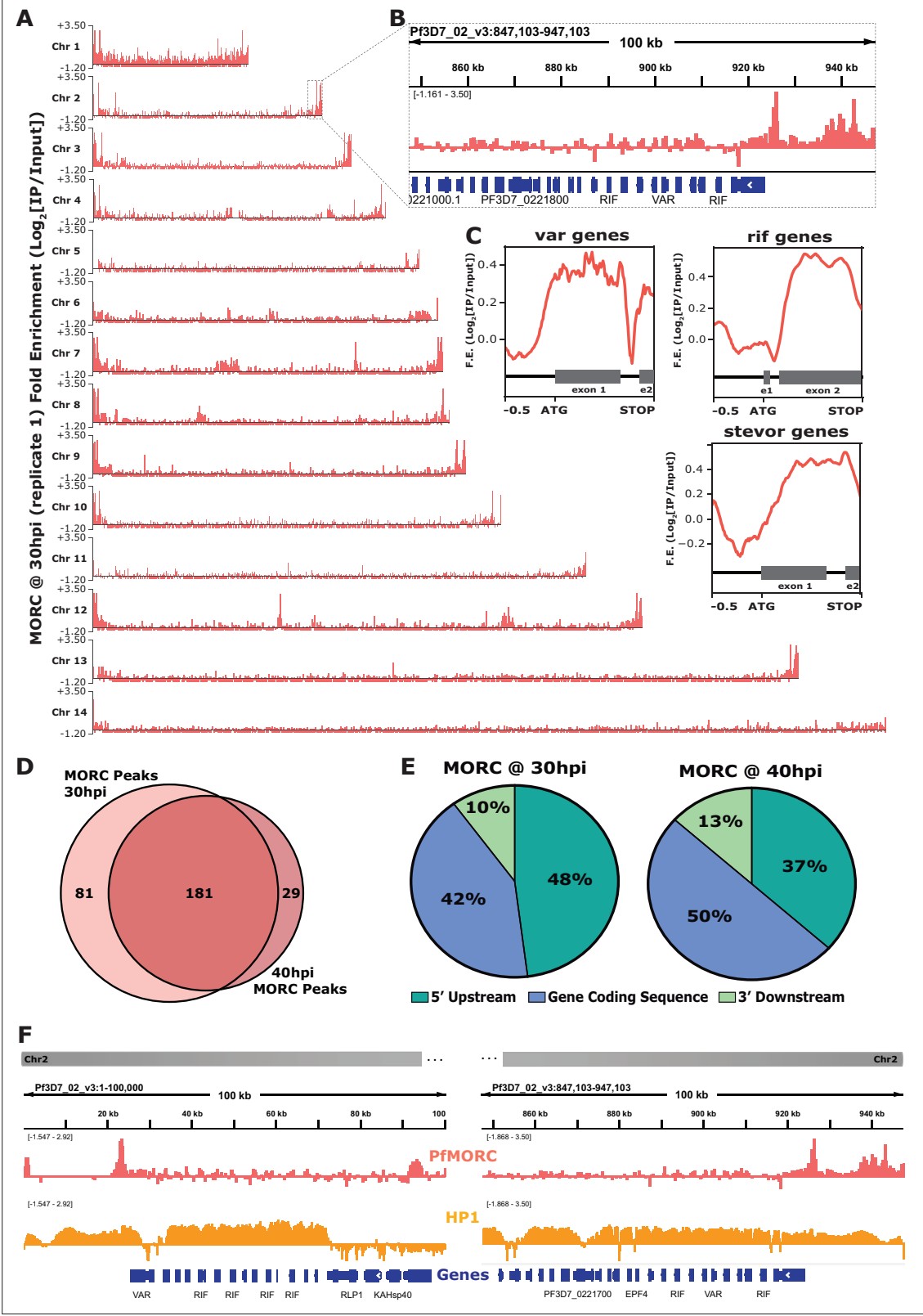

**Figure 2.** Genome-wide occupancy of *Pf*MORC reveals localization to hypervariable surface antigen genes at 30 hr and 40 hr. (**A**) Coverage tracks of *Pf*MORC across all 14 *P. falciparum* chromosomes. Plotted values are fold enrichment (Log2[IP/Input]) of a representative replicate at 30 hr. (**B**) Zoom-in of the last 100 kb region of chromosome two from (**A**). Gene annotations represented in blue bars (*P. falciparum* 3D7 strain, version 3, release 57; PlasmoDB.org). (**C**) Mean fold enrichment of *Pf*MORC occupancy across all *var* genes (top left), all *rif* genes (top right), and all *stevor* genes (bottom

*Figure 2 continued on next page*

*Figure 2 continued*

right), excluding pseudogenes. Graphical representation of exons to scale for each gene family annotated below enrichment plot in grey (e1 = exon one; e2 = exon two). (**D**) Quantitative Venn diagram comparing the number of MACS2 called peaks across each timepoint (light pink for 30 hr; dark pink for 40 hr). (**E**) Pie charts showing the type of genomic locations *Pf*MORC peaks overlap at both 30 hr and 40 hr. Pink slices are 5` regions upstream of the ATG start site of genes, blue slices are coding sequences/gene bodies of genes, and green slices are 3` regions downstream of the stop codon of genes. (**F**) Zoom-in of the first 100 kb region (left) and the last 100 kb region (right) of chromosome two. Plotted are the ChIP-seq fold enrichment of *Pf*MORC (top track; pink) and heterochromatin protein 1 (HP1; middle track; orange) with gene annotations (bottom track; blue bars; *P. falciparum* 3D7 strain, version 3, release 57; PlasmoDB.org).

The online version of this article includes the following figure supplement(s) for figure 2:

**Figure supplement 1.** Comparision of ChiP-seq enriched peaks across different *Pf*MORC samples.

**Figure supplement 2.** ChIP-seq profiling of *Pf*MORC fold enrichment across var gene regions.

**Figure supplement 3.** ChIP-seq profiling of *Pf*MORC fold enrichment across *rif* gene regions.

**Figure supplement 4.** ChIP-seq profiling of *Pf*MORC fold enrichment across stevor gene regions.

**Figure supplement 5.** The heatmaps show the transcript abundance (*Chappell et al., 2020*) of putative *Pf*MORC gene targets at 30 hr (**A**) and 40 hr (**B**).

motifs at both the 30 hpi and 40 hpi timepoints, which suggests that more than one sequence-specific transcription factor may be responsible for recruiting *Pf*MORC to specific genomic regions (*Figure 3—figure supplement 2*). The common motifs identified across replicates and timepoints are RGTGCAW or TGCACACA, both of which are similar or identical to the in vitro and/or in vivo DNA-binding motif of *Pf*AP2-I (RGTGCAW) or PF3D7_0420300 (TGCACACA), respectively, suggesting that these ApiAP2 factors may play major roles in *Pf*MORC recruitment (*Figure 3—figure supplement 2*).

In addition to overlapping occupancy with ApiAP2 TFs, we found that *Pf*MORC co-localizes with the depletion of H3K36me2 (*Figure 3C and D*), which is demarcated by the SET2 methyltransferase (*Jiang et al., 2013*), both at 30 hpi and 40h pi. No other significant co-localization was found between *Pf*MORC and other epigenetic marks (H2A.Z, H3K9ac, H3K4me3, H3K27ac, H3K18ac, H3K9me3, H3K36me2/3, H4K20me3, and H3K4me1) (*Figure 3—figure supplement 3*), suggesting it has a unique binding preference not shared with other heterochromatin markers. Therefore, it is likely that *Pf*MORC co-localizes with other, as yet uncharacterized, epigenetic marks. In summary, *Pf*MORC was found to be recruited to 5'-untranslated regions (UTRs), gene body regions, and subtelomeric regions of repressed, multigene families, and overlaps with other known ApiAP2 binding sites and DNA motifs.

## Depletion of *Pf*MORC results in the upregulation of late-stage genes associated with invasion

*Pf*MORC association with chromatin remodelers has been shown (*Bryant et al., 2020*), but how *Pf*MORC regulates gene expression in the asexual stage has not been evaluated. In *T. gondii*, the TgMORC:TgApiAP2 complex acts as a transcriptional repressor of sexual commitment (*Farhat et al., 2020*; *Srivastava et al., 2023*; *Antunes et al., 2024*). Here, we found that *Pf*MORC co-immunopre-cipitates with several chromatin remodeling proteins and many ApiAP2 transcription factors. Further-more, our ChIP-seq data revealed that *Pf*MORC is located at subtelomeric regions of the genome. Based on this evidence, we hypothesized that *Pf*MORC may regulate transcriptional changes during blood stage development of the parasite. To knock down the expression of *Pf*MORC (*Pf*MORC-KD), sorbitol-synchronized MORC^HA-glmS parasites (22–24 hpi) were subjected to 2.5 mM glucosamine (GlcN) treatment for little over 48 hr when they reached the trophozoite stage (32 hpi ±3 hpi), at which point parasites were harvested for RNA isolation for transcriptomic analysis. In parallel, another flask with *Pf*MORC^HA-glmS parasites was set up without GlcN and used as control for RNA-seq comparison. We previously demonstrated that treatment with 2.5 mM GlcN results in a 50% knockdown of *Pf*MORC protein but does not cause any growth delay; using >2.5 mM GlcN caused a measurable slow growth and reduced parasitemia (*Singh et al., 2021b*). Three biological replicates with and without 2.5 mM GlcN were collected for knockdown transcriptomics to ensure reproducibility.

For each *Pf*MORC RNA-seq sample, gene counts were used to identify the differentially expressed genes (DEGs). Significant threshold parameters were assigned to a p-value<0.05, yielding a total of 2558 DEGs (*Supplementary file 6*). Applying a $\log_2$-fold change cutoff from >1 to <-1 and filtering out pseudogenes reduced this number to 163 DEGs. Among these, 60 genes display more abun-dant transcripts, whereas 103 genes were reduced relative to control parasites grown without GlcN

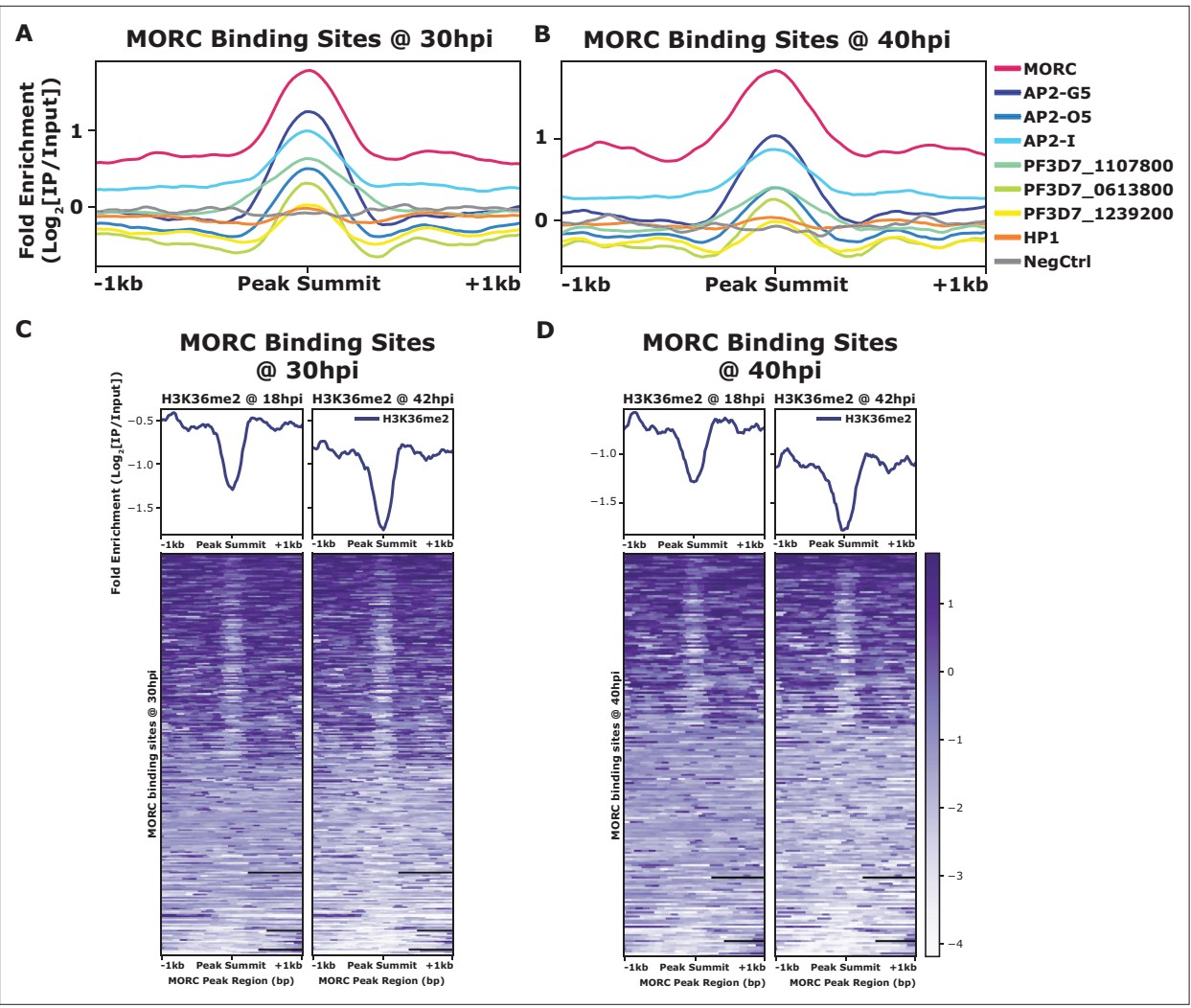

**Figure 3.** Comparision of mean fold enrichment of PfMORC with ApiAP2 transcription factors and othe epigenetic markers at different time points. (**A**) Mean fold enrichment (Log2[IP/Input]) of PfMORC, six associated factors (AP2-G5, AP2-O5, AP2-I, PF3D7_1107800, PF3D7_0613800, and PF3D7_1239200), HP1, and a negative no-epitope control across PfMORC binding sites at the 30 hr timepoint. (**B**) Mean fold enrichment (Log2[IP/Input]) of PfMORC, six associated factors (AP2-G5, AP2-O5, AP2-I, PF3D7_1107800, PF3D7_0613800, and PF3D7_1239200), HP1, and a negative no-epitope control across PfMORC binding sites at the 40 hr timepoint. (**C**) Mean fold enrichment (Log2[IP/Input]) and heatmap of two H3K36me2 epigenetic mark timepoints across PfMORC binding sites at 30 hr. (**D**) Mean fold enrichment (Log2[IP/Input]) and heatmap of two H3K36me2 epigenetic mark timepoints across PfMORC binding sites at 40 hr.

The online version of this article includes the following figure supplement(s) for figure 3:

**Figure supplement 1.** Overlap between PfMORC and other ApiAP2 transcription factors binding regions.

**Figure supplement 2.** DNA motif analyses from these different categories: (1) unique to 30 hpi ChIP-seq timepoint, (2) 30 hpi ChIP-seq timepoint, (3) overlap of ChIP-seq timepoint, (4) 40 hpi ChIP-seq timepoint, and (5) unique to 30 hpi ChIP-seq timepoint.

**Figure supplement 3.** Mean fold enrichment (Log2[IP/Input]) summary plot (top) and full heatmap (bottom) of fold enrichment of 10 selected epigenetic marks (H2A.Z, H3K9ac, H3K4me3, H3K27ac, H3K18ac, H3K9me3, H3K36me2/3, H4K20me3, and H3K4me1) across PfMORC binding sites at the 30 hr and 40 hr timepoints.

(**Figure 4A**). Pathway and functional enrichment analysis yielded several genes from apical organelles. GO analysis revealed gene clusters enriched with molecular function (p-adj=0.0006) involved in the movement into the host environment and entry into the host, and molecular function of protein binding (p-adj=0.009) (**Figure 4B**, **Supplementary file 7**). More specifically, upregulated genes implicated in the invasion of merozoites were found to be expressed prematurely upon PfMORC KD; these include several rhoptry-associated genes, notably PfRON2 (PF3D7_1452000) and PfRON3 (PF3D7_1252100). Both PfRON2 and PfRON3 are part of a micronemal complex at the erythrocyte

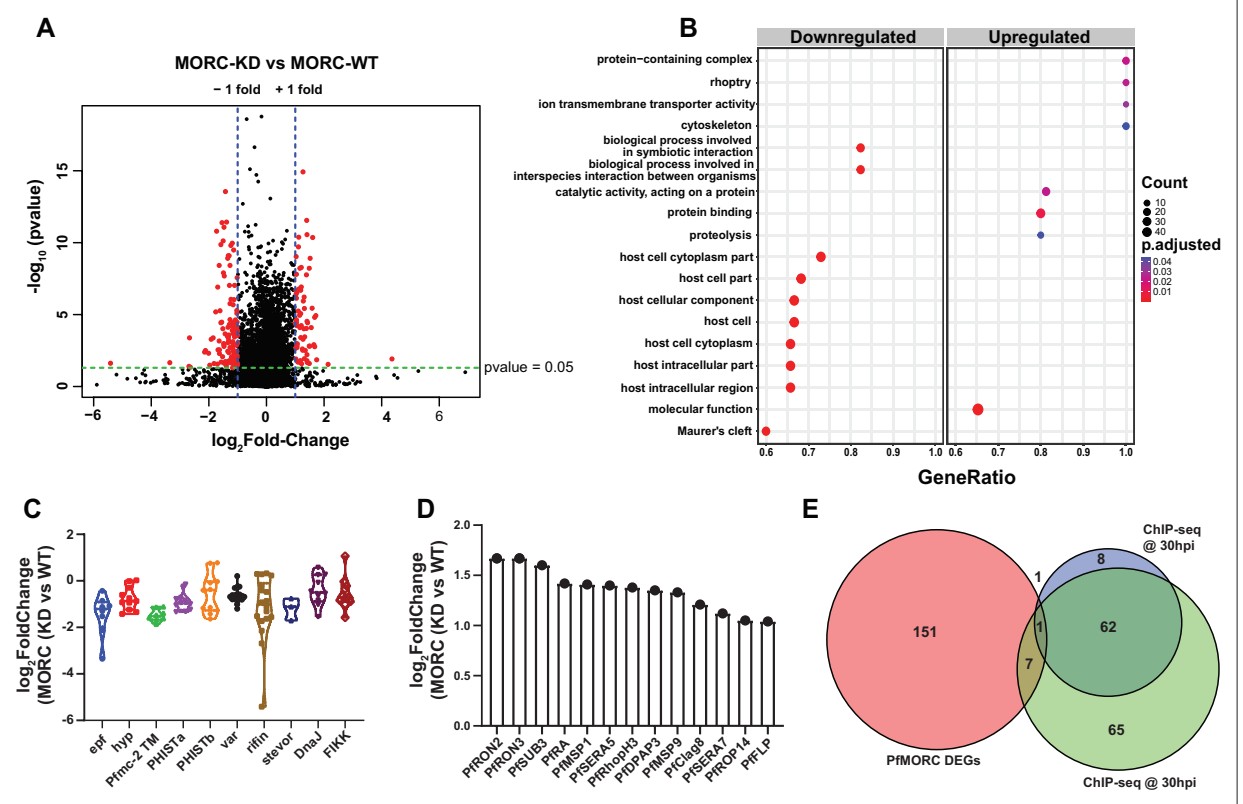

**Figure 4.** Transcriptome analysis of *Pf*MORC knockdown revealed differential gene expression. (**A**) Volcano plot displaying the differential gene expression in *Pf*MORC-KD compared to the *Pf*MORC-WT phenotype. Tightly synchronized *Pf*MORC^HA-glmS parasites (32 hpi ± 3 hr) were split into two populations, one of which was treated with 2.5 mM GlcN to obtain the *Pf*MORC knockdown phenotype and the other was not treated with GlcN to obtain wild-type phenotype. Total RNA-seq was performed, and significant threshold parameters for differentially expressed genes (DEGs) were assigned to a p-value <0.05 and -log$_2$ fold change >1 from three biological replicates. (**B**) Scatter plot shows upregulated and downregulated DEGs which were further categorized for pathway and functional enrichment analysis using the KEGG database (p-adjusted value<0.05). The circle size at the vertical axis represents the number of genes in the enriched pathways and the horizontal axis represents gene richness as a ratio of DEGs in the pathways to the total genes in a specific pathway. (**C**) The violin plot of log$_2$ fold change of genes belonging to the multigene family is constructed from *Pf*MORC-KD vs. *Pf*MORC-WT, which shows DEGs of multigene family proteins upon *Pf*MORC knockdown. (**D**) The bar plot illustrates the upregulated DEGs of apical organelle origin in *Pf*MORC-KD parasites involved in host cell invasion. (**E**) Venn diagram showing the comparison between genes obtained from ChIP-seq data and DEGs obtained from RNA-seq data. Both 30hpi and 40 hpi timepoints were taken for comparison and showed high overlap with each other but there was no overlap with RNA-seq data.

The online version of this article includes the following figure supplement(s) for figure 4:

**Figure supplement 1.** Comparison of transcriptional changes with melatonin treatment.

membrane where *Pf*RON2 anchors *Pf*AMA1 to facilitate merozoite invasion (*Srinivasan et al., 2013*). In addition, *Pf*SUB3 (PF3D7_0507200), *Pf*SERA5 (PF3D7_0207600), and *Pf*DPAP3 (PF3D7_0404700), all of which are critical for schizont rupture (*Yeoh et al., 2007*; *Arastu-Kapur et al., 2008*), were among the upregulated DEGs (*Figure 4C*). In general, we found that depletion of *Pf*MORC leads to the upregulation of invasion-related genes, which suggests that *Pf*MORC has an additional function in the regulation of genes specifically associated with RBC invasion.

### *Pf*MORC-depleted parasites downregulate hypervariable gene families

Among the genes with reduced mRNA abundance, DEGs linked to cytoadherence, antigenic variation, and interaction with host (*Figure 4B and D*) were over-represented in the GO analysis. Many of the genes enriched in the downregulated group belong to the clonally variant *var* multigene family that represents 60 members encoding the *P. falciparum* erythrocyte membrane proteins (*Pf*EMP1), which upon switching expression aid in pathogenesis and immune evasion (*Guizetti and Scherf, 2013*). Furthermore, a cluster of genes encoding putative exported proteins were also enriched

including members of the exported protein family (EPFs), *Plasmodium* exported protein (hyp), and *Plasmodium* exported protein (PHISTa/b). Other significantly overrepresented downregulated genes belonged to serine/threonine protein kinases, the FIKK family (*Ward et al., 2004*), most of which are exported to the RBC, and Maurer's clefts two transmembrane proteins (Pfmc-2TM) (*Figure 4D*). Notably, expressed proteins are conserved across the *Plasmodium* family and remain confined to the hypervariable subtelomeric region of *P. falciparum* chromosomes (*Sargeant et al., 2006*).

## Comparison of ChIP-seq gene targets and DEGs by RNA-seq

To identify whether genes found to be dysregulated after *Pf*MORC knockdown are associated with the genome-wide occupancy of *Pf*MORC, we correlated the gene targets identified by ChIP-seq with the DEGs determined by RNA-seq. We identified a total of 135 gene targets from the 30 hpi ChIP-seq timepoint, 72 gene targets from the 40 hpi ChIP-seq timepoint, and 163 DEGs by RNA-seq. The low correlation between the ChIP-seq gene targets and the RNA-seq DEGs suggests that *Pf*MORC genome-wide occupancy is more likely involved in chromatin structure, rather than specific regulation of gene targets (*Figure 4E*). Likely, *Pf*MORC localizes to these sites to aid in chromatin condensation, as shown in other eukaryotic systems (*Zhang et al., 2019*; *Zhong et al., 2023*).

## Discussion

Periodic gene expression during the asexual blood stage directly corresponds to the timing in which gene products are needed (*Bozdech et al., 2003*) and shows oscillation patterns associated with circadian rhythms (*Smith et al., 2020*; *Subudhi et al., 2020*). Transcription factors are critical regulators of this dynamic pattern in concert with epigenetic regulators and genome-wide changes to chromatin structure (*Painter et al., 2011*; *Toenhake et al., 2018*; *Jeninga et al., 2019*; *Hollin et al., 2021*). The ApiAP2 family members display unique binding preferences in the genome (*Campbell et al., 2010*), undergo stage-specific expression (*Painter et al., 2011*), and have been identified to regulate virtually all stages of development across multiple *Plasmodium* species (*Jeninga et al., 2019*). To date, ApiAP2 proteins have been reported in transcriptional silencing of clonally variant genes, regulation of invasion genes, and sexual commitment, acting through interaction with other epigenetic factors, such as heterochromatin protein 1 [*Pf*HP1 (*Flueck et al., 2010*; *Brancucci et al., 2014*; *Fraschka et al., 2018*)], bromodomain protein 1 [*Pf*BDP1 (*Santos et al., 2017*; *Josling et al., 2020*; *Quinn et al., 2022*)], or general control non-depressible 5 [*Pf*GCN5 (*Miao et al., 2021*)]. Despite the known DNA-binding sites of many ApiAP2 proteins (*Flueck et al., 2010*; *Santos et al., 2017*; *Sierra-Miranda et al., 2017*; *Josling et al., 2020*; *Carrington et al., 2021*; *Shang et al., 2021b*, *Singh et al., 2021a*, *Russell et al., 2022*; *Shang et al., 2022*; *Bonnell et al., 2023*), limited information is available for other accessory proteins. Recently, the *T. gondii* ortholog *Tg*MORC was identified in a complex with HDAC1, AP2XII-2, and AP2XII-1:AP2XI-2and orchestrates epigenetic rewiring of sexual gene transcription (*Farhat et al., 2020*; *Srivastava et al., 2023*; *Antunes et al., 2024*). In *P. falciparum*, *Pf*MORC has been copurified with several ApiAP2 proteins, as shown by different independent studies (*Hillier et al., 2019*; *Bryant et al., 2020*; *Singh et al., 2021b*).

In this study, we used an integrated multi-omics approach to explore the function of *Pf*MORC during asexual blood stage development. Using immunoaffinity-based purification, we identified a number of nuclear proteins that interact with *Pf*MORC. More specifically, *Pf*MORC was associated with *Pf*AP2-G5, which is essential for gametocytogenesis (*Shang et al., 2021b*), *Pf*AP2-I, which is required for the expression of many invasion-related genes (*Santos et al., 2017*), and with other ApiAP2 TFs of unknown function (*Pf*AP2-O5, PF3D7_1107800, PF3D7_0613800, PF3D7_0420300, and PF3D7_1239200). The identification of *Pf*SW1 and *Pf*CHD1 in association with *Pf*MORC strengthens the link between *Pf*MORC and chromatin remodeling. This finding suggests that *Pf*MORC may participate in regulating chromatin structure and gene expression during the IDC, notably the specific regulation of *var* genes. We note that some nuclear proteins were not identified in this study despite their documented interaction with *Pf*MORC in other studies (*Hillier et al., 2019*; *Bryant et al., 2020*; *Singh et al., 2021a*, *Subudhi et al., 2023*). This may be due to differences in experimental conditions between the various studies and to the low abundance of the proteins. Despite this, our identification of several nuclear proteins in complex with *Pf*MORC provides insights into potential interactions and

regulatory mechanisms underlying gene regulation and chromatin remodeling in *P. falciparum*; this may have implications for developing new strategies to combat malaria.

In other eukaryotes, MORC proteins function in gene repression and chromatin compaction at heterochromatic regions (**Koch et al., 2017**). Therefore, to determine if *Pf*MORC localizes to heterochromatic regions across the *P. falciparum* genome, we performed ChIP-seq assays with *Pf*MORC^HA-glmS parasites during peak *Pf*MORC expression (trophozoite and schizont stages). In general, *Pf*MORC occupied regions 5'-upstream of the ATG start site or was bound within coding region, irrespective of the developmental stage. Most importantly, *Pf*MORC peaks were reproducibly detected in subtelomeric regions containing hypervariable multigene families, including the *var* genes, consistent with the findings that *Pf*MORC localized to *var* gene promoters as reported using dCas9 (**Bryant et al., 2020**). The specific binding pattern of *Pf*MORC near or within *Pf*HP1-bound regions suggests two related critical functions: heterochromatin condensation and gene repression. It is possible that *Pf*MORC changes the nucleosome landscape either by direct association or by binding to different chromatin remodelers. Recent work in *Arabidopsis* has further confirmed the importance of *At*MORC paralogs in gene regulation by chromatin compaction (**Zhong et al., 2023**), which resembles the function of *Pf*MORC in *P. falciparum*. Interestingly, in *T. gondii*, the major function of *Tg*MORC was in the repression of sex determination genes (**Farhat et al., 2020**), suggesting MORC family proteins have the capacity to perform diverse functions across eukaryotic organisms. Future studies should determine if *Pf*MORC depletion similarly influences the rate of sexual commitment to *P. falciparum* gametocytogenesis, which is known to be regulated by epigenetic mechanisms. The role of *Pf*HP1 has been shown in regulating sexual commitment by repressing *Pf*AP2-G and virulence genes expression (**Brancucci et al., 2014**), whereas gametocyte development 1 (*Pf*GDV1) displaces *Pf*HP1 itself and induces asexual developing parasites to sexual differentiation (**Filarsky et al., 2018**).

We also compared *Pf*MORC occupancy with available ChIP-seq data for the interacting ApiAP2 proteins. Interestingly, our analysis revealed a significant overlap between the binding sites of all ChIP-ed ApiAP2 proteins with *Pf*MORC, suggesting a cooperative role in gene regulation. We also identified enriched motifs similar to those bound by our ApiAP2 proteins of interest, further suggesting the functional cooperation between *Pf*MORC and ApiAP2 proteins. Of note, only one of the seven ApiAP2 of interest in our study has not been ChIP-ed to date (PF3D7_0420300) (**Shang et al., 2022**). However, we found an enrichment of the DNA motif (TGCACACA) at *Pf*MORC-bound sites. This motif is bound in vitro by PF3D7_0420300 (**Bonnell et al., 2023**), suggesting that this ApiAP2 protein may localize to these regions. Overall, since these seven ApiAP2 proteins are expressed at distinct timepoints during the *P. falciparum* cycle (**Bozdech et al., 2003**) and regulate different sets of genes (**Santos et al., 2017**; **Josling et al., 2020**; **Shang et al., 2021b**, **Shang et al., 2022**; **Subudhi et al., 2023**), we believe this indicates a variety of functions for *Pf*MORC at different stages. In addition, a recent study in *Arabidopsis* reported that *At*MORC-mediated regulation of transcription may be due to both direct chromatin interactions and indirect association via sequence-specific transcription factors (**Zhong et al., 2023**) consistent with a complex landscape of chromatin remodeling by MORC proteins. We also interrogated the co-localization of *Pf*MORC and numerous epigenetic profiles (H2A.Z, H3K9ac, H3K4me3, H3K27ac, H3K18ac, H3K9me3, H3K36me2/3, H4K20me3, and H3K4me1), with a specific focus on H3K36me2, since this histone modification has been shown to act as a global repressive effector in *P. falciparum* (**Karmodiya et al., 2015**) and other organisms (**Strahl et al., 2002**; **Wagner and Carpenter, 2012**). Interestingly, we did not find a strong co-localization with any of these epigenetic marks, suggesting a unique role of the epigenetic landscape on *Pf*MORC global occupancy. Therefore, the functional association between *Pf*MORC and a specific, unknown epigenetic mark remains to be determined.

Before this work, there was no direct evidence correlating *Pf*MORC-mediated transcriptional changes during the IDC of *P. falciparum*. Therefore, we investigated the DEGs in *Pf*MORC knockdown parasites and found two distinct subsets of enriched genes. The upregulated DEGs were enriched with genes related to invasion, while the downregulated DEGs belong to hypervariable genes associated with parasite virulence. It suggests that *Pf*MORC has very dynamic function across *Plasmodium* asexual cycle as we also identified *Pf*AP2-I in CoIPed proteins, which is shown to regulate invasive gene transcription (**Santos et al., 2017**). Overall, our data with *Pf*MORC-KD revealed a change in the expression of both antigenically variable and invasion-related genes. Expression of clonally variant genes occurs in a different but tightly regulated manner in IDC (**Scherf et al., 1998**; **Scherf et al.,

*2008*), which, in the light of our data, may be regulated by *Pf*MORC occupancy. Data regarding changes in *var* gene expression are difficult to interpret, as the KD experiments were performed on a parasite that had not been recloned and may therefore express more than one *var* gene. Single-cell experiments would be needed to clarify the effect of *Pf*MORC KD on *var* gene regulation. We were surprised by the small number of DEGs detected upon knockdown of *Pf*MORC, which we believe may be due to incomplete knockdown.

We previously showed that parasites in which *Pf*MORC is knocked down display reduced sensitivity to melatonin (*Singh et al., 2021a*). This prompted us to investigate if overall gene expression in *Pf*MORC-KD parasites is affected by melatonin treatment. We detected only very slight changes (*Figure 4—figure supplement 1*), suggesting that melatonin does not exert its effect on gene expression through *Pf*MORC, or at least that the latter protein plays a minor role in the process.

Overall, this study shows that *Pf*MORC interacts with different ApiAP2 TFs, in line with previous studies (*Hillier et al., 2019*; *Bryant et al., 2020*; *Singh et al., 2021a*, *Subudhi et al., 2023*) and a recently published parallel study (*Chahine et al., 2023*). Furthermore, we found that *Pf*MORC is localized at sub-telomeric regions and contains significant overlap with the binding sites of several ApiAP2 transcription factors. Our results support a role for *Pf*MORC in the regulation of hypervariable genes that are essential for *Plasmodium* virulence and rhoptry genes critical to RBC invasion. Collectively, our data identify an important role for *Pf*MORC in chromatin organization, as well as in the epigenetic regulation of gene expression through regulatory complexes with an array of transcription factors, making it an attractive drug target.

## Materials and methods

### *Plasmodium falciparum* culture

The *P. falciparum* 3D7 strain (BEI Resources, MRA-102) and MORC constructs (3D7 background) were cultured at 37°C in RPMI 1640 medium supplemented with 0.5% Albumax II (Gibco) (*Trager and Jensen, 1976*). Parasites were grown under a 5% $CO_2$, 5% $O_2$, and 90% $N_2$ atmosphere. Cultures were synchronized with 5% sorbitol (*Lambros and Vanderberg, 1979*). Parasites were tested negative for mycoplasma contamination using PCR.

### Coimmunoprecipitation and mass spectrometry

Infected erythrocytes at the trophozoite stage were collected from culture and washed twice in 1× phosphate-buffered saline (PBS). The culture pellet was suspended in PBS containing 0.05% (w/v) saponin to lyse the erythrocyte membrane and centrifuged at 8000 × *g* for 10 min. The supernatant was discarded, and the parasite pellet was washed three times in cold PBS. To perform the co-immunoprecipitation, we followed the manufacturer's protocol (ChromoTek, gta-20). Samples were lysed in modified RIPA buffer (50 mM Tris, pH 7.5, 150 mM NaCl, 0.5% sodium deoxycholate, 1% Nonidet P-40, 10 µg/ml aprotinin, 10 µg/ml leupeptin, 10 µg/ml, 1 mM phenylmethylsulfonyl fluoride, benzamidine) for 30 min on ice. The lysate was precleared with 50 µl of protein A/G-Agarose beads at 4°C for 1 hr and clarified by centrifugation at 10,000 × *g* for 10 min. The precleared lysate was incubated overnight with an anti-GFP-Trap-A beads (ChromoTek, gta-20) antibody. The magnetic beads were then pelleted using a magnet (Invitrogen), and the beads were washed extensively using wash buffer (50 mM Tris, pH 7.5, 150 mM NaCl, 0.5% sodium deoxycholate, 1% Nonidet P-40) to minimize the detection of non-specific proteins. To elute the immunoprecipitated proteins, the magnetic beads were resuspended in 2× SDS loading buffer and resolved by SDS-PAGE. Following SDS-PAGE, the whole gel band for each sample was excised from three independent experiments and further analyzed by mass spectrometry. We used a service provider (CEFAP core-facility de Espectometria de Massa) to analyze GFP-coimmunoprecipitated proteins.

### In-gel digestion and peptide desalting

Protein bands from polyacrylamide gels were cut into pieces (approximately 1 mm$^3$), transferred to a clean 1.5 ml low binding tube and washed with washing solution (40% acetonitrile, 50 mM ammonium bicarbonate) until the bands were completely distained, and dehydrated with ACN 100% for 5 min

followed by vacuum centrifugation. Proteins were then reduced with 10 mM dithiothreitol in 50 mM ammonium bicarbonate and incubated for 45 min at 56°C. Protein alkylation was performed with 55 mM iodoacetamide in 50 mM ammonium bicarbonate and incubated for 30 min at room temperature. Proteins were digested into peptides by trypsin overnight reaction at 37°C. The trypsin reaction was stopped with 10% TFA (1% TFA final concentration). The supernatant was collected into a new tube. Extraction buffer (40% ACN/0.1% TFA) was added to the gel pieces and incubated for 15 min on a thermomixer at room temperature. The supernatant was transferred to the same microtube. The peptide extraction was performed twice and then dried in a vacuum centrifuge. Peptides were resuspended in 0.1% TFA for desalting.

## Nano LC-MS/MS analysis

The LC-MS/MS system employed was an Easy-nano LC 1200 system (Thermo Fisher Scientific Corp) coupled to an Orbitrap Fusion Lumos mass spectrometer equipped with a nanospray source (Thermo Fisher Scientific Corp). Samples were loaded onto a trapping column (Acclaim PepMap 0.075 mm, 2 cm, C18, 3 µm, 100 A; Thermo Fisher Scientific Corp.) in line with a nano-LC column (Acclaim PepMap RSLC 0.050 mm, 15 cm, C18, 2 µm, 100 A; Thermo Fisher Scientific Corp.). The gradient was 5–28% solvent B (A 0.1% FA; B 90% ACN, 0.1% FA) for 25 min, 28–40% B for 3 min, 40–95% B for 2 min, and 95% B for 12 min at 300 nl/min. Orbitrap Fusion Lumos mass spectrometer operated in positive mode. The full MS scan had an automatic gain control (AGC) of $5 \times 10^5$ ions and a maximum filling time of 50 ms. Each MS scan was acquired at 120K full width half maximum high resolution in the Orbitrap with a mass range of m/z 400–1600 Da. High-resolution dissociation with a normalized collision energy set at 30 was used for fragmentation. The resulting MS/MS fragment ions were detected in the Orbitrap mass analyzer at a resolution of 30,000. An AGC of $5 \times 10^4$ ions and a maximum injection time of 54 ms were used. All raw data were accessed in Xcalibur software (Thermo Scientific).

## Database searches and bioinformatics analyses

Raw files were imported into MaxQuant version 1.6.17.0 for protein identification and quantification. For protein identification in MaxQuant, the database search engine Andromeda was used against the UniProt *P. falciparum* 3D7 strain (March 2021, 5388 entries release). The following parameters were used: carbamidomethylation of cysteine (57.021464 Da) as a fixed modification, oxidation of methionine (15.994915 Da), and N-terminal acetylation protein (42.010565 Da) were selected as variable modifications. Enzyme specificity was set to full trypsin with a maximum of two missed cleavages. The minimum peptide length was set to seven amino acids. For label-free quantification, the 'match between runs' feature in MaxQuant was used, which is able to identify the transfer between samples based on the retention time and accurate mass, with a 0.7 min match time window and 20 min alignment time window. Label-free quantification was performed using MaxQuant software with the 'match between run' and iBAQ features activated. Protein LFQ and iBAQ ratios were calculated for the two conditions, and the protein IDs were divided accordingly. The mass spectrometry proteomics data have been deposited to the ProteomeXchange Consortium via the PRIDE (*Perez-Riverol et al., 2022*) partner repository with the dataset identifier PXD036092. PlasmoDB database was used to perform GO analysis.

## Total RNA extraction and RNAseq

Infected RBCs containing tightly synchronized trophozoite stage parasites (32 hpi ± 3 hr) were harvested and resuspended in TRIzol (Thermo Fisher Scientific). Total RNA was extracted from three independent experiments following the manufacturer's protocol, and an RNA cleanup kit (QIAGEN) was used to achieve high-purity RNA. The isolated RNA was quantified using a NanoDrop ND-1000 UV/Vis spectrophotometer, and RNA quality was determined using an RNA ScreenTape System (Agilent 2200 TapeStation). Total RNA (10,000 ng) from each sample was stabilized in RNAStable (Biomatrica) and sent to the Micromon Genomics facility at Monash University for next-generation sequencing.

RNA samples were prepared using the MGITech RNA Directional Library Prep Kit V2 (Item No. 1000006385), as per the manufacturer's instructions, with the following parameters: input RNA: 50 ng, fragmentation: target of 200–400 bp, 87°C for 6 mins, adapter clean-up: 200–400 bp and

library amplification cycles: 16. Libraries were assessed for quantity using the Invitrogen Qubit and dsDNA HS chemistry (Item No. Q33230) and quality/quality using the Agilent Fragment Analyzer 5200 with the HS NGS Fragment Kit (Item No. DNF-473–0500). The libraries were pooled in equimolar concentrations and sequenced using an MGITech DNBSEQ-G400RS sequencing instrument with High-Throughput Sequencing Set (FCL PE100, Item No. 1000016950) chemistry.

The quality of the RNA-seq libraries was evaluated using the FastQC tool. Next, we used Salmon (V1.9.0) (*Patro et al., 2017*) quant with default arguments to quantify the expression of all transcripts in the PlasmoDB release-58 Pfalciparum3D7 genome. The transcript expression was summarized to gene-level expression with tximport 1.22.0 (*Soneson et al., 2015*). Finally, the gene counts were used to detect DEGs with DESeq2 (1.34.0) (*Love et al., 2014*). Furthermore, only genes with >1 $\log_2$ fold change and adjusted p-value<0.1 were considered significant for further analysis.

## Chromatin immunoprecipitation followed by high-throughput sequencing (ChIP-seq)

*Pf*MORC ChIP-seq was performed similarly to previously published ChIP-seq experiments in *P. falciparum* (*Josling et al., 2020*; *Singh et al., 2021a*, *Russell et al., 2022*). Five samples in total were collected: biological duplicates using the *Pf*MORC[HA-glmS] parasite line at the trophozoite stage (30 hpi), biological duplicates using the *Pf*MORC[HA] parasite line at the schizont stage (40 hpi), and a single replicate using the *Pf*MORC[GFP] parasite line at the schizont stage (40 hpi). In brief, the protocol includes five steps: (1) treated with 1% formaldehyde to crosslink the suspended *Pf*MORC[HA-glmS] (or *Pf*MORC[GFP]) parasite cultures (at least $10^8$ trophozoite- or schizont-stage parasites synchronized with 10% sorbitol more than one cycle prior) at 37°C for 10 min; (2) collected parasite nuclei using prechilled glass Dounce homogenizer for 100 strokes per $10^9$ trophozoites/schizonts; (3) lysed parasite nuclei and mechanically sheared chromatin until sufficiently sheared using Covaris Focus-Ultrasonicator M220 (5% duty cycle, 75 W peak incident power, 200 cycles per burst, 7°C, for 5 min). (4) We collected 10% of each sample for the non-immunoprecipitated 'input' control and then immunoprecipitated the remaining 90% of each sample. The remaining 90% of each sample was immunoprecipitated with 1:1000 anti-HA antibody (0.5 mg/ml Roche Rat Anti-HA High Affinity [11867423001]) or 1:1000 anti-GFP antibody (0.1 mg/ml Abcam Rabbit Anti-GFP [Ab290]) overnight at 4°C with rotation. (5) DNA was purified after reverse crosslinking using a MinElute column (QIAGEN) as directed and quantified by a Qubit fluorometer (Invitrogen).

## ChIP-seq library prep for Illumina sequencing

The *Pf*MORC ChIP-seq libraries were performed similarly to previously published ChIP-seq experiments in *P. falciparum* (*Josling et al., 2020*; *Singh et al., 2021b*, *Russell et al., 2022*). DNA sequencing libraries were prepared for high-throughput Illumina sequencing on the NextSeq 2000 with the 150 × 150 single-end mode. The library prep underwent 12 rounds of amplification using KAPA HiFi polymerase. Completed libraries were quantified using the Qubit fluorometer (Invitrogen) for high-sensitivity DNA and library sequence length by the Agilent TapeStation 4150.

## ChIP-seq data analysis and peak calling

The *Pf*MORC ChIP-seq datasets were analyzed similar to previously published ChIP-seq experiments in *P. falciparum* (*Josling et al., 2020*; *Singh et al., 2021b*, *Russell et al., 2022*). Raw sequencing reads were trimmed (Trimmomatic v0.32.3) to remove Illumina adaptor sequences and low-quality reads below 30 Phred (SLIDINGWINDOW: 4:30). FastQC (v0.11.9) was used to check the quality after trimming. Processed reads were then mapped to the *P. falciparum* genome (release 57) using BWA-MEM (v0.7.17.2) simple Illumina mode with multiple mapped reads filtered out (MAPQ = 1). Once the sequences were mapped, MACS2 (v2.2.7.1) was used to call peaks with each biological replicate and its paired input sample using a standard significance cutoff (q-value = 0.01). Using BedTools Multiple Intersect (v2.29.2), the narrow peak output file for each replicate was overlapped to identify the significant peaks in the overlap between both ChIP-seq biological replicates. The overlapping regions were then used to identify an enriched DNA motif (STREME Meme Suite: *Bailey, 2021*), and putative gene targets of *Pf*MORC were defined as genes with peaks within 2 kb upstream of the gene target ATG start codon or peaks within gene bodies. In a situation with any peaks between gene targets in a head-to-head orientation, the closest gene was chosen.

## Acknowledgements

This work was supported by grants from Fundação de Amparo a Pesquisa de São Paulo (FAPESP) to CRSG (2017/08684-7 and 2018/07177-7) and to MKS (2019/09490-7). CRSG is supported by 'bolsa de produtividade' by CNPq. This work was supported by the National Institutes of Health grant number R01-AI125565 to ML and T32-GM125592 to VAB. We are grateful to Haemocentro Hospital do Servidor Público for providing blood and plasma. Work in CD laboratory is supported by RMIT University and by grant APP2003712 from the Australian Health and Medical Research Council (NHMRC). We thank Prof. Paolo Di Mascio and Graziella E Ronsein for the Mass spectrometry analyses performed at the Redox Proteomics Core of the Mass Spectrometry Resource at Chemistry Institute, University of São Paulo (FAPESP 2012/12663-1, 2016/00696-3, 2023/00995-4, CEPID Redoxoma 2013/07937-8), and Dr. Mariana P Massafera for her technical assistance.

## Additional information

### Funding

| Funder | Grant reference number | Author |
| --- | --- | --- |
| Fundação de Amparo à Pesquisa do Estado de São Paulo | 2017/08684-7 | Celia RS Garcia |
| Fundação de Amparo à Pesquisa do Estado de São Paulo | 2018/07177-7 | Celia RS Garcia |
| Fundação de Amparo à Pesquisa do Estado de São Paulo | 2019/09490-7 | Maneesh Kumar Singh |
| National Institutes of Health | R01-AI125565 | Manuel Llinas |
| National Institutes of Health | T32-GM125592 | Victoria Ann Bonnell |
| National Health and Medical Research Council | APP2003712 | Christian Doerig |

The funders had no role in study design, data collection and interpretation, or the decision to submit the work for publication.

### Author contributions

Maneesh Kumar Singh, Conceptualization, Data curation, Formal analysis, Investigation, Methodology, Writing – original draft, Writing – review and editing; Victoria Ann Bonnell, Conceptualization, Data curation, Methodology, Writing – original draft; Israel Tojal Da Silva, Formal analysis; Verônica Feijoli Santiago, Investigation, Methodology; Miriam Santos Moraes, Methodology; Jack Adderley, Resources; Christian Doerig, Conceptualization, Resources, Supervision, Project administration, Writing – review and editing; Giuseppe Palmisano, Supervision, Project administration, Writing – review and editing; Manuel Llinas, Conceptualization, Supervision, Funding acquisition, Project administration, Writing – review and editing; Celia RS Garcia, Conceptualization, Resources, Supervision, Funding acquisition, Project administration, Writing – review and editing

### Author ORCIDs

Maneesh Kumar Singh ⓘ https://orcid.org/0000-0003-1474-4695
Israel Tojal Da Silva ⓘ https://orcid.org/0000-0002-4687-1499
Verônica Feijoli Santiago ⓘ https://orcid.org/0000-0002-0052-9532
Christian Doerig ⓘ https://orcid.org/0000-0002-3188-094X
Giuseppe Palmisano ⓘ https://orcid.org/0000-0003-1336-6151
Manuel Llinas ⓘ https://orcid.org/0000-0002-6173-5882
Celia RS Garcia ⓘ https://orcid.org/0000-0003-2825-1701

Reviewer #1 (Public Review): https://doi.org/10.7554/eLife.92201.3.sa1
Author response https://doi.org/10.7554/eLife.92201.3.sa2

## Additional files

### Supplementary files

• Supplementary file 1. Intersecting peptide hits from three different experiments obtained from *Pf*MORC-GFP parasite lysate.

• Supplementary file 2. Complete list of potential *Pf*MORC interacting proteins enriched in CoIP eluates and identified in LC–MS/MS from three independent experiments and fold change ≥1.5× GFP/3D7.

• Supplementary file 3. Gene Ontology (GO) terms for proteins identified in *Pf*MORC-GFP lysate and normalized to fold change >1.5.

• Supplementary file 4. Gene IDs and annotations for gene targets of *Pf*MORC at 30hpi and 40 hpi determined by ChIP-seq genome-wide occupancy. Putative gene targets of *Pf*MORC were defined as genes with peaks within 2 kb upstream of the gene target ATG start codon or peaks within gene bodies. In a situation with any peaks between gene targets in a head-to-head orientation, the closest gene was chosen.

• Supplementary file 5. Gene Ontology (GO) terms for gene targets of *Pf*MORC at 30 hpi and 40 hpi determined by ChIP-seq genome-wide occupancy. GO Terms were defined using PlasmoDB.org GO Term enrichment function (Biological Process with p-value cutoff of 0.05).

• Supplementary file 6. Full list of differentially expressed genes in *Pf*MORC-KD vs. *Pf*MORC-WT.

• Supplementary file 7. Gene Ontology (GO) terms for differentially expressed genes identified in *Pf*MORC-KD/*Pf*MORC-WT RNAseq.

• Supplementary file 8. List of differentially expressed genes in *Pf*MORC-KD vs. *Pf*MORC-WT after 100 nM melatonin treatment for 5 hr.

• MDAR checklist

### Data availability

The proteomics data have been deposited to the ProteomeXchange Consortium via the PRIDE partner repository with the dataset identifier PXD044256. All ChIP-seq samples were submitted to GEO under accession code GSE239393. All RNA-seq samples were submitted to GEO under accession code GSE241313.

The following datasets were generated:

| Author(s) | Year | Dataset title | Dataset URL | Database and Identifier |
|---|---|---|---|---|
| Santiago VF, Garcia CRS | 2024 | *Plasmodium falciparum* MORC protein modulates gene expression through interaction with heterochromatin | https://proteomecentral.proteomexchange.org/cgi/GetDataset?ID=PXD044256 | ProteomeXchange, PXD044256 |
| Singh MK, Bonnell VA, Tojal da Silva I, Santiago VF, Moraes MS, Adderley J, Doerig C, Llinás M, Garcia CRG | 2024 | *Pf*MORC modulates gene expression through interactions with heterochromatin in *Plasmodium falciparum* | https://www.ncbi.nlm.nih.gov/geo/query/acc.cgi?&acc=GSE239393 | NCBI Gene Expression Omnibus, GSE239393 |
| Singh M, Tojal da Silva I, Adderley J, Doerig C, da Silva Garcia C | 2023 | *Plasmodium falciparum* MORC protein modulates gene expression through interaction with heterochromatin | https://www.ncbi.nlm.nih.gov/geo/query/acc.cgi?acc=GSE241313 | NCBI Gene Expression Omnibus, GSE241313 |

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
