## [Editor Report · eLife assessment]

This study provides **valuable** insights into how chromatin-bound *Pf*MORC controls gene expression in the asexual blood stage of *Plasmodium falciparum*. By interacting with key nuclear proteins, *Pf*MORC is predicted to affect expression of genes relating to host invasion and variable subtelomeric gene families. Correlating transcriptomic data with in vivo chromatin analysis, the study provides **convincing** evidence for the role of *Pf*MORC in epigenetic transcriptional regulation.

---

## [Referee Report · Reviewer #1 (Public Review)]

Summary:

The study provides valuable insights into the role of PfMORC in Plasmodium's epigenetic regulation, backed by a comprehensive methodological approach. The overarching goal was to understand the role of PfMORC in epigenetic regulation during asexual blood stage development, particularly its interactions with ApiAP2 TFs and its potential involvement in the regulation of genes vital for Plasmodium virulence. To achieve this, they conducted various analyses. These include a proteomic analysis to identify nuclear proteins interacting with PfMORC, a study to determine the genome-wide localization of PfMORC at multiple developmental stages, and a transcriptomic analysis in PfMORCHA-glmS knockdown parasites. Taken together, this study suggests that PfMORC is involved in chromatin assemblies that contribute to the epigenetic modulation of transcription during the asexual blood stage development.

Strengths:

The study employed a multi-faceted approach, combining proteomic, genomic, and transcriptomic analyses, providing a holistic view of PfMORC's role. The proteomic analysis successfully identified several nuclear proteins that may interact with PfMORC. The genome-wide localization offered valuable insights into PfMORC's function, especially its predominant recruitment to subtelomeric regions. The results align with previous findings on PfMORC's interaction with ApiAP2 TFs. Notably, the authors meticulously contextualized their findings with prior research adding credibility to their work.

Weaknesses:

While the study identifies potential interacting partners and loci of binding, direct functional outcomes of these interactions remain an inference. The use of the glmS ribozyme system to achieve a 50% reduction in PfMORC transcript levels makes it difficult to understand the role of PfMORC solely in terms of chromatin architecture without considering its impact on gene expression. Although assessing the overall impact of acute MORC depletion was beyond the scope of the study, it would have been informative.

---

## [Author Response]

The following is the authors’ response to the original reviews.

**eLife assessment**
This study provides valuable insights into how chromatin-bound PfMORC controls gene expression in the asexual blood stage of *Plasmodium falciparum*. By interacting with key nuclear proteins, PfMORC appears to affect expression of genes relating to host invasion and subtelomeric var genes. Correlating transcriptomic data with in vivo chromatin insights, the study provides solid evidence for the central role of PfMORC in epigenetic transcriptional regulation through modulation of chromatin compaction.
**Public Reviews:**

**Reviewer #1 (Public Review):**
Summary:The study provides valuable insights into the role of PfMORC in Plasmodium's epigenetic regulation, backed by a comprehensive methodological approach. The overarching goal was to understand the role of PfMORC in epigenetic regulation during asexual blood stage development, particularly its interactions with ApiAP2 TFs and its potential involvement in the regulation of genes vital for Plasmodium virulence. To achieve this, they conducted various analyses. These include a proteomic analysis to identify nuclear proteins interacting with PfMORC, a study to determine the genome-wide localization of PfMORC at multiple developmental stages, and a transcriptomic analysis in PfMORCHA-glmS knockdown parasites. Taken together, this study suggests that PfMORC is involved in chromatin assemblies that contribute to the epigenetic modulation of transcription during the asexual blood stage development.Strengths:The study employed a multi-faceted approach, combining proteomic, genomic, and transcriptomic analyses, providing a holistic view of PfMORC's role. The proteomic analysis successfully identified several nuclear proteins that may interact with PfMORC. The genome-wide localization offered valuable insights into PfMORC's function, especially its predominant recruitment to subtelomeric regions. The results align with previous findings on PfMORC's interaction with ApiAP2 TFs. Notably, the authors meticulously contextualized their findings with prior research, including pre-prints, adding credibility to their work.Weaknesses:While the study identifies potential interacting partners and loci of binding, direct functional outcomes of these interactions remain an inference. The authors heavily rely on past research for some of their claims. While it strengthens some assertions, it might indicate a lack of direct evidence in the current study for particular aspects. The declaration that PfMORC may serve as an attractive drug target is substantial. While the data suggests its involvement in essential processes, further studies are required to validate its feasibility as a drug target.
**Reviewer #2 (Public Review):**
Summary:This is a paper entitled "*Plasmodium falciparum* MORC protein modulates gene expression through interaction with heterochromatin" describes the role of PfMORC during the intra-erythrocytic cycle of *Plasmodium falciparum*. Garcia et al. investigated the PfMORC-interacting proteins and PfMORC genomic distribution in trophozoites and schizonts. They also examined the transcriptome of the parasites after partial knockdown of the transcript.Strengths:This study is a significant advance in the knowledge of the role of PfMORC in heterochromatin assembly. It provides an in-depth analysis of the PfMORC genomic localization and its correlation with other chromatin marks and ApiAP2 transcription factor binding.Weaknesses:However, most of the conclusions are based on the function of interacting proteins and the genomic localization of the protein. The authors did not investigate the direct effects of PfMORC depletion on heterochromatin marks. Furthermore, the results of the transcriptomic analysis are puzzling as 50% of the transcripts are downregulated, a phenotype not expected for a heterochromatin marker.
**Recommendations for the authors:**

**Reviewer #1 (Recommendations For The Authors):**
Suggestions for improved or additional experiments, data, or analyses.• Figure 1A and Table 1: the authors should incorporate a volcano plot in their proteomic results presentation. This graphical representation can provide a more intuitive grasp of the most relevant proteins associated with PfMORC in terms of both their abundance and significance. It will aid in swiftly pinpointing proteins with the most notable differential associations. This will complement the comprehensive overview provided by the authors, referencing past research where PfMORC was detailed.

We thank the reviewer for the suggestion. We agree with the reviewer that the volcano plot we now provide does indeed bring comprehensive information on associations between PfMORC and other cellular proteins. The volcano plot presented in the revised manuscript as Figure 1A, was generated using the normalized MS/MS counts from the anti-GFP and 3D7 (control) proteomics datasets (n=3). The potential PfMORC interacting proteins were determined using the fold changes and p-values between the two datasets, as provided in Table 1.

Several protein interactors were strongly supported by statistical analysis (p-value), while others showed weaker p-value due to variability between replicates. Indeed, the total number of proteins identified in the three replicates, shown in the Venn diagram (Supplemental Figure 1D), exhibits a good overlap between the replicates but a lower number of identified proteins in the GFP-E1 sample. This variability was observed also in the statistical analysis. Indeed, by analyzing the GFP/3D7 ratios, some proteins have a significant difference in abundance (fold change greater than 1.5x) in one of the groups but do not meet the statistical threshold. For more clarity, we have included the -log p-value for the proteins listed in Table 1.

Overall, these results demonstrate that many ApiAP2 proteins and several chromatin-associated factors interact with PfMORC.

• Given the plethora of proteins detected in the PfMORC eluate, it raises the question of how many are genuine MORC interactors versus those that are merely nearby molecules acting adjacently. These might incidentally end up in the immunoprecipitate due to unintended interactions with DNA or chromatin. While the M&M section mentions that the beads were thoroughly washed, there is no specification about the washing buffer or its stringency (i.e., salinity level). At higher salinities, one could isolate core complexes of interactors associated with DNA or even RNA carryover.

We apologize for this omission and have now added the buffer composition used to wash the beads. This section now reads "To perform the co-immunoprecipitation we followed the manufacturer protocol (ChromoTek, gta-20). Samples were lysed in modified RIPA buffer (50 mM Tris, pH 7.5, 150 mM NaCl, 0.5% sodium deoxycholate, 1% Nonidet P-40, 10 µg/ml aprotinin, 10 µg/ml leupeptin, 10 µg/ml, 1 mM phenylmethylsulfonyl fluoride, benzamidine) for 30 min on ice. The lysate was precleared with 50 µl of protein A/G-Agarose beads at 4°C for 1 h and clarified by centrifugation at 10,000 × g for 10 min. The precleared lysate was incubated overnight with an anti-GFP antibody using anti-GFP-Trap-A beads (ChromoTek, gta-20). The magnetic beads were then pelleted using a magnet (Invitrogen) and washed 3 times with wash buffer (10 mM Tris/Cl pH 7.5, 150 mM NaCl, 0.05 % Nonidet P40 Substitute, 0.5 mM EDTA)."

We used the same salt concentration for immunoprecipitation as was used in the lysis buffer to minimize the binding of non-specific proteins. The wash buffer composition is updated in the revised manuscript. The immunoprecipitations were done in biological triplicates to ensure reproducibility and statistical support. A number of proteins are common across all three replicates. We also used wild-type parasites (non-GFP) as a negative control to eliminate non-specific hits, and we used a log2-fold change ≥1.5 relative to wild type parasites as our cutoff between the comparison groups.

We believe that these conditions provide the stringency required to identify high confidence PfMORC interacting proteins, although this still leaves a possibility for additional lower affinity interactions. Future studies will certainly follow up candidate interaction partners to better define this complex. However, the complexity of the complex resembles that reported previously in *Toxoplasma gondii* (Farhat et al. 2020, Nat Microbiol) as well another report on the PfMORC complexes: https://elifesciences.org/reviewed-prepri nts/92499

• The authors demonstrate that PfMORC creates distinct peaks in and around HP1-bound areas (Figure 2F), hinting at a specific role for PfMORC in heterochromatin compaction, boundary definition, and gene silencing. This pattern is clearly depicted in an example in Figure 2F. It would be beneficial to know if this enrichment profile is replicated elsewhere and, if so, it would be worthwhile to quantify it.

This is an excellent point. Yes, this pattern is seen across the entire genome, where PfMORC is apposed to PfHP1-bound heterochromatic regions. As indicated in the manuscript, we have quantified this effect genome-wide; however, since we already display compiled data for Chromosome 2 (at both chromosome ends) pertaining to the position of PfMORC relative to PfHP1 we do not feel it is essential to provide such a figure for the entire genome as it does not alter the central message of our manuscript. Figure 2F is representative of the genome-wide distribution of PfMORC relative to PfHP1. The raw genome-wide data are available in Supplementary Information for further inspection of specific loci on other chromosomes.

Recommendations for improving the writing and presentation.MAIN TEXTPanel e, referenced both in the main text and legend, is missing from Figure 4. This missing panel represents a significant finding of the study, highlighting according to the authors a low correlation between ChIP-seq gene targets and RNA-seq DEGs. This observation implies that PfMORC's global occupancy is more aligned with shaping chromatin architecture than directly regulating specific gene targets. In light of this, the authors should rephrase parts of their manuscript (including abstract and title) to avoid suggesting that PfMORC acts primarily (directly) as a gene regulator, emphasizing instead its role in influencing the topological structure of chromosomes.

We have modified the title as suggested by the reviewer to more accurately reflect that PfMORC modulates chromatin architecture rather than acting as a direct regulator of specific genes. Our new title is: A *Plasmodium falciparum* MORC protein complex modulates epigenetic control of gene expression through interaction with heterochromatin

We apologize for the omission of Figure 4e, which is now included in the revised manuscript. We found PfMORC occupancy on all chromosomes at subtelomeric regions, which are known to harbor genes related to immune evasion and antigenic variation (including most of the var genes). This study is also in agreement with Bryant et al. (PMID 32816370) which reported PfMORC occupancy along with PfISW1 at var gene promoters. PfMORC has also been identified in complexes with various ApiAP2 proteins in a proteome-wide study (Hillier et al. Cell Rep, PMID 31390575), as well as in immunoprecipitations of PfAP2-G2 (Singh et al., Mol Micro, PMID 33368818) and PfAP2-P (Subudhi et al., Nat Microbiol, PMID 37884813). The recent study by Subudhi et al. reports that PfAP2-P is involved in the regulation of var gene expression, antigenic variation, trophozoite development and parasite egress. It is therefore possible that PfMORC may have different effects on transcriptional regulation through interactions with different ApiAP2 transcription factors. Our comparison of PfMORC with known ApiAP2 protein occupancy reveals a high level of overlap, indicating that PfMORC may affect gene expression in various ways throughout the asexual cycle. Additionally, Hillier et al. show that PfMORC interaction is not limited to ApiAP2 but also implicates several other chromatin remodellers, which is consistent with our own results. We do not imply direct regulation of transcription via PfMORC in our manuscript. To the contrary, we suggest that it interacts with heterochromatin and thereby plays a role in the epigenetic control of asexual blood stage transcriptional regulation which is also clarified in the revised abstract.

Another limitation of differential gene expression was use of the glmS ribozyme system, which resulted in only 50% depletion of the PfMORC transcript. There may still be enough PfMORC to rescue the gene expression we could not detect correctly. Therefore, it is challenging to interpret the function of PfMORC in only chromatin architecture but not in gene expression.If we believe that PfMORC in Plasmodium isn't mainly adjusting gene expression, the authors' suggestion that MORC is targeted by some AP2s becomes puzzling. How do we make sense of these different ideas? The authors need to clarify this to maintain consistency in their findings.

Based on our data, we hypothesize that PfMORC acts as an accessory protein for ApiAP2 transcription factors. In a number of studies, including ours and the concurrent publication in eLife (https://elifesciences.org/reviewed-preprints/92499), PfMORC co-IPed with several ApiAP2 proteins, suggest it has multiple functions. In our previous study we showed that PfMORC expression is highest in mid and late asexual stages. A comparison of the PfMORC occupancy with 6 ApiAP2 (having different expression profile) suggest plasticity in PfMORC function. We have revised our discussion to make this hypothesis more transparent for the readers.

The authors should cite Farhat et al. 2020 (Extended Data Fig. 1a), as it similarly identified 3 different ELM2-containing proteins in Toxoplasma MORC-associated complexes. This previous work provides context and supports the observations made with PfMORC in this study.

Thank you for the suggestion and pointing out this omission. We have indeed cited the work of the Farhat group in the original manuscript and have now included this additional reference to corroborate the text and provide further support to our conclusions.

Minor corrections to the text and figures.• Panel e is missing from Figure 4.

As mentioned above Panel e is now included in Figure 4.

• The captions are very minimally detailed. An effort must be made to better describe the panels as well as which statistical tests were used. As it stands, this is not really up to standard.

We have elaborated the captions with more detailed descriptions, and we now provide additional information where further clarification was necessary.

**Reviewer #2 (Recommendations For The Authors):**
The study lacks a direct correlation between the inferred function of PfMORC and the heterochromatin state of the genome after its depletion. It would be interesting to perform chip-seq on known heterochromatin markers such as H3K9me3, HP1 or H3K36me2/3 to measure the consequences of PfMORC depletion on global heterochromatin and its boundaries.

While the proposed experiments are certainly interesting, they are beyond the scope of this study. The current manuscript is focused on PfMORC occupancy, its interacting partners, and its impact on differential gene regulation after PfMORC depletion in asexual parasites. Nonetheless, we did in fact compared the PfMORC occupancy with that of various heterochromatin markers (H2A.Z, H3K9ac, H3K4me3, H3K27ac, H3K18ac, H3K9me3, H3K36me2/3, H4K20me3, and H3K4me1) at 30hpi and 4hpi time points. These data are presented in Supplemental Figure 9. We did not find any significant colocalization, but documented the presence of PMORC in H3K36me2 depleted regions.

The PfMORC depletion was performed using a glms-based genetic system and the reviewer did not find any quantification of the depletion level at 24h or 36h. This is particularly important as the authors present RNA-seq data at these time points.

We would like to clarify that RNA-seq was performed on 32hpi parasites after approximately 48 h treatment with 2.5 mM GlcN. At the trophozoite and schizont stage, PfMORC expression is high, which is why we selected these time points for RNA-seq (32hpi) and ChIP-seq (30hpi and 40hpi). PfMORC protein expression after GlcN treatment is analyzed in our previous paper (Singh et al., Sci Rep, PMID 33479315), where treatment with 2.5 mM GlcN leads to 50% reduction in PfMORC transcript at 32hpi. This is referenced in the Results section; we decided not to repeat the same experiment in the current manuscript.

The authors performed a thorough analysis of the correlations between ApiAP2 binding, histone modification and genomic localization of PfMORC (their chip-seq data). However, they found an inverse relationship between H3K36me2, a known histone repressive mark, and PfMORC genomic localization. This is particularly surprising when PfMORC itself is presented as a heterochromatin marker. The wording of this data is confusing in the results section (lines 257-258) and never discussed further. This important data should at least be discussed to make sense of this apparent contradiction.

H3K36me2 indeed acts as a global repressive mark in *P. falciparum*. However, our hypothesis implies that PfMORC not only overlaps with H3K36me2 depleted region, but also interacts with other epigenetic regulators. Therefore, we propose that PfMORC is part of chromatin remodeling complexes involved in heterochromatin dynamics. Moreover, we did not see any overlap between several other heterochromatin markers, suggesting it has a unique binding preference not shared with other heterochromatin markers. Based on this study and parallel work submitted by Chahine et al. (https://elifesciences.org/reviewed-preprints/92499#abstract), it is evident that PfMORC is crucial for gene regulation and chromatin structure maintenance as shown in other organisms. Currently, we do not know what the apparent mutual exclusion between H3K36me2 and PfMORC implies mechanistically or how PfMORC interaction with heterochromatin aids in chromatin integrity. In *Arabidopsis thaliana*, MORC binding leads to chromatin compaction and reduces DNA accessibility to transcription factors, thereby repressing gene expression. In *P. falciparum*, overlap in the binding region of PfMORC with different transcription factors suggests several possibilities that require further investigation. Since there is only one gene encoding a PfMORC protein in *P. falciparum*, it is possible that PfMORC function is not limited to chromatin integrity, but it may also function to modulate gene expression at different stages. To fully explore the function of PfMORC will require investigating the functional role of the other interacting partners we and others have identified.

We have modified the result section per the reviewer's suggestion, and we now also discuss this finding in more detail in the discussion section.

The ChIP-seq data are central to this manuscript. However, the presentation of this data in Figure 2A suggests that it is very noisy (particularly for Chr1). It would be of interest to present the called peaks together with the normalized data so that the reader can assess the quality of the ChIP-seq data.

Our results clearly demonstrate the enrichment of PfMORC in sub-telomeric regions and internal heterochromatic islands. These results are consistent across all of our replicates taken at two independent time points of parasite asexual blood stage development and correlate well with the results of Le Roch: https://elifesciences.org/reviewed-preprints/92499. The raw data files have been provided and can be re-analyzed by any user.

The RNA-seq data showed that only a few genes are affected after 24 h of PfMORC depletion. Furthermore, there is an equal number of up- and down-regulated genes. It is not clear why depletion of a heterochromatin marker would induce down-regulation of genes. How these data relate to the partial depletion of PfMORC is not discussed.

We would like to clarify that RNA-seq experiment was performed at 32hpi after GlcN following knockdown as previously described (Singh et al., Sci Rep, PMID 33479315). Briefly, synchronous, early trophozoites stage (24hpi) PfMORCglmS-HA parasites were treated with 2.5 mM GlcN until they reached the trophozoite stage (32 hpi) in the next cycle. These parasites were then collected for analysis by RNA-seq. We did not detect a substantial log-fold change at this point because only 50% of the transcripts were depleted in the glmS-based PfMORC knockdown system. However, we have seen a distinctive pattern of up (60) and down (103) regulated DEGs that are comprised of egress-related genes or surface antigens. We believe that PfMORC interacts with different ApiAP2 proteins, as shown in Figure 3A, and consequently exhibits multiple functions. This finding has now been corroborated in several other recent studies (See response to Reviewer 1 above).